# One Size Does Not Fit All: A Multifaceted Approach to Educate Families about Newborn Screening

**DOI:** 10.3390/ijns10030044

**Published:** 2024-06-26

**Authors:** Marianna H. Raia, Molly M. Lynch, Alyson C. Ward, Jill A. Brown, Natasha F. Bonhomme, Vicki L. Hunting

**Affiliations:** 1Expecting Health, Washington, DC 20016, USA; nbonhomme@expectinghealth.org (N.F.B.); vhunting@expectinghealth.org (V.L.H.); 2RTI International, Research Triangle Park, NC 27709, USA; mlynch@rti.org (M.M.L.); jillbrown@rti.org (J.A.B.); 3National Center for Hearing Assessment and Management (NCHAM), Utah State University, 2615 Old Main Hill, Logan, UT 84322, USA; alyson.ward@usu.edu

**Keywords:** newborn screening, family engagement, quality improvement, family partners in healthcare, maternal and child health

## Abstract

All families deserve access to readily available, accurate, and relevant information to help them navigate the newborn screening system. Current practices, limited resources, and a siloed newborn screening system create numerous challenges for both providers and families to implement educational opportunities to engage families in ways that meet their needs with relevant and meaningful approaches. Engaging families in newborn screening, especially those from historically underserved communities, is necessary to increase knowledge and confidence which leads to overall improved outcomes for families. This article describes three strategies that the Navigate Newborn Screening Program developed, tested, and implemented in the United States, including online learning modules, a prenatal education pilot program, and social media awareness campaign, as well as the extent to which they were successful in reaching and educating families about newborn screening. Using quality improvement methods and evidence-driven approaches, each of these three strategies demonstrate promising practices for advancing awareness, knowledge, and self-efficacy for families navigating the newborn screening system—particularly families in medically underserved and underrepresented communities. A model for bidirectional engagement of families is outlined to support scaling and implementing promising educational efforts for both providers and families in the newborn screening system.

## 1. Introduction

Newborn screening (NBS) benefits babies by detecting life-threatening or life-altering conditions before symptoms negatively impact the child, and is arguably one of the most effective public health programs [1,2,3]. Despite newborn screening being demonstrably successful in life-saving detection and intervention, persistent barriers to newborn screening knowledge remain, and are often worsened by evolving practice, policy, and social views on public health [4]. This underscores the need for novel, adaptable and evidence-based approaches to educate and engage families.

Numerous reports have shown the benefits and positive outcomes of engaging families as partners in systems of healthcare and service delivery, including improved health outcomes, patient safety, and legislative policies established to reduce healthcare costs [4,5,6,7]. However, systematic data on family and provider experiences in NBS are lacking, leaving a gap in fully understanding the barriers that families face when learning about the newborn screening system. Engaging families should include early involvement in program design, development, and implementation with the goal of leveraging lived experiences to improve processes and outcomes [5,8]. Many families are not made aware of newborn screening prior to or at the time of their child’s birth, making it challenging for them to understand the importance of and possible outcomes of NBS, including follow-up testing. Complex health systems, siloed infrastructure, and limited resources exacerbate existing knowledge gaps around newborn screening and are often magnified among less educated and medically underserved populations, with research noting that infants of less educated parents are less likely to receive timely diagnosis and services [9].

All families deserve access to readily available, accurate, and appropriate information to help them navigate the newborn screening system. There is substantial evidence to support the ideal time to provide parents with information about NBS being during pregnancy [10,11,12,13]. Receiving information about NBS prenatally and from a healthcare provider are both factors associated with parent satisfaction with that information [10]. Furthermore, parents participating in interviews indicated that NBS education should begin before labor and delivery, and recommended that it be provided either in brochure or video format during the prenatal period [11]. In another study, participants indicated that the ideal time for NBS education was during the third trimester of pregnancy [12]. Even though the benefits of prenatal education about newborn screening are acknowledged [13], this education is not common practice. Specifically, a study of prenatal care providers in California found that only 33% of them discussed newborn screening with their patients [14]. While pregnancy is a critical time point for information, additional opportunities for education remain, including birth, time of screening, screen positive results, diagnosis, and follow-up. Recognizing the nuanced and individualized needs for families and those providing information to families at each of these unique time points in the newborn screening journey is essential to developing meaningful education practices. 

Systems of care should empower families and provide opportunities to be involved in the development of systems to ensure that their needs and those of newborns, infants, and children are addressed [15]. The Newborn Screening Family Education Program (funded by the United States (U.S.) Health Resources and Services Administration (HRSA)/Maternal and Child Health Bureau (MCHB) to Expecting Health at Genetic Alliance) focused on increasing education around newborn screening amongst families in the United States. With support from partner organizations, outputs of the program were bundled under the title Navigate Newborn Screening (Navigate NBS) to increase education around newborn screening among families in the U.S., specifically families that are typically underserved and/or underrepresented. 

The mission of the Navigate NBS Program is to develop opportunities for all individuals to learn about newborn screening and to create educational and training resources that build confidence in families in becoming leaders in the NBS system [16]. To measure awareness and knowledge of newborn screening and to define how increasing NBS knowledge may improve families’ experience with newborn screening, the Navigate NBS Program tested a variety of strategies. This article describes three strategies that the program developed and tested, including online learning modules, a prenatal education pilot program, and social media awareness campaign, and the extent to which they were successful in reaching and educating families about NBS.

## 2. Materials and Methods

Navigate NBS staff utilized the Model for Improvement (MFI), developed by Associates in Process Improvement [17] as the framework to develop and improve promising strategies to reach and educate U.S.-based families. The MFI is often used in healthcare and public healthcare settings to accelerate process and outcome improvement while mitigating human and capital costs. The model has two parts: it starts with three fundamental questions: (1) What are we trying to accomplish? (2) How will we know a change is an improvement? and (3) What changes can we make that will result in improvement? These are followed by the second part: the Plan–Do–Study–Act (PDSA) cycles, designed to test changes on a small scale and incrementally build scale as confidence increases that the change is indeed resulting in improvement. This methodology was applied to identify, develop, and test a series of strategies aimed at raising awareness and knowledge of newborn screening.

The strategies implemented through Navigate NBS followed Expecting Health/Baby’s First Test’s Newborn Screening Education Best Practices Framework, using the elements of (1) identifying the goal of educational effort, (2) deciding on the population that will be prioritized, and (3) when and how the education will be implemented [18]. To identify potential needs and strategies for educating families about newborn screening, the Navigate NBS program conducted a nationwide needs assessment of 819 U.S.-based parents, expecting parents, individuals with NBS conditions, or family members of individuals with NBS conditions, with focus on those living inside of medically underserved areas (MUAs). As previously published, this assessment highlighted that individuals from MUAs reported less awareness of NBS and do not receive NBS education at the optimal time before birth, which could indicate that they experience inequities in NBS education [9]. In addition, the results from the needs assessment indicated that online, self-paced learning from trusted sources is the preferred communication and learning method for most families, particularly those that are harder to reach and from MUA. This information informed the development, testing, and implementation of the Navigate NBS online learning modules. In collaboration with families and professional leaders in NBS, the modules were designed to help a wide range of families build knowledge about NBS and self-efficacy to participate in NBS systems at the individual, state, and federal levels.

Using these frameworks, the Navigate NBS team developed and tested three strategies to reach and educate families about NBS in different ways. Numerous educational materials were developed in partnership with newborn screening experts including families, U.S.-based newborn screening laboratorians, follow-up educators, researchers, evaluators, community health clinics, public health professionals, physicians, genetic counselors, and bioethicists. Materials developed included online, video-based training modules, educational booklets in both English and Spanish, and tailored evaluation instruments. Details of each of these materials are described in subsequent sections and are publicly available at www.expectinghealth.org (accessed on 11 June 2024).

### 2.1. Strategy 1: Online Education Modules

The program applied quality improvement methods to identify opportunities for improvement and subsequently developed additional training modules to expand the accessibility, reach, and engagement of families. A focus group including families and clinicians was conducted to test the modules and provide initial feedback for improvement prior to a broad launch. Participants that tested the modules were recruited through national advocacy and family organizations, as well as U.S. state/territory newborn screening partners. The Navigate NBS staff designed and launched a partner toolkit to support more meaningful and targeted outreach. Social media graphics, downloadable one-pagers, infographics, and customized links to the program’s online learning tools were shared broadly through national conferences, websites, and community partners. Enrolled participants were families with diverse newborn screening conditions, geographic locations, and level of engagement with the newborn screening community. These participants provided valuable feedback that encouraged the Navigate NBS staff to revise modules, add modules, and develop best ways to disseminate the modules. 

Through iterative feedback from participants, the initial Navigate NBS online curriculum, Navigate Newborn Screening, included both core and optional modules, including the following: Section 1: The Newborn Screening Process;Section 2: Newborn Screening Results;Section 3: Types of Conditions Detected;Section 4: Questions to Ask Your Healthcare Provider;Section 5: How to Tell Your Newborn Screening Story;Optional Section 1: Available Resources and Tools;Optional Section 2: Using Social Media for Advocacy.

Learning was assessed through checkpoint questions at the conclusion of each section. As part of program evaluation, we asked participants to complete a form about changes in knowledge and self-efficacy related to NBS based on participation in the educational modules. Using their Likert scale responses, we summed the number of participants who agreed or strongly agreed with these statements: 

After taking the educational module…

I feel more knowledgeable about newborn screening.I feel more confident to find information about NBS.I feel more confident to talk with a healthcare provider about NBS.I feel more confident to advocate for myself or my child.I feel more confident to serve as a leader in the community.

In an effort to reach and educate more families, Navigate NBS also released additional curriculums in 2022, including Navegando por la Evaluación del Recién Nacido, a version of the training designed for Spanish-speaking families, and Navigate newborn Screening Quick Bites, an abbreviated version of the original online curriculum. Similar to the English version, a Spanish-speaking test group was established, and community focus groups were facilitated to assess for cultural and language competencies. 

### 2.2. Strategy 2: Prenatal Education in Medically Underserved Clinics

Recognizing that pregnancy is a preferred time for mothers to receive information about NBS [10], Navigate NBS staff developed an educational initiative designed to support sharing information and increasing knowledge of NBS prior to delivery. To increase awareness and to reach medically underserved expectant mothers, Navigate NBS staff developed, tested, and implemented an initiative providing NBS education in two community-based obstetrics and gynecology clinics using an informational “flip book”. The initiative began in June 2021 and was piloted in a high-risk obstetrics clinic in Houston, Texas that served primarily Spanish-speaking pregnant women. To scale this project, Navigate NBS subsequently modified and tested the flip book with a second pilot site in a community health clinic that engaged midwives who support an Amish/Mennonite (Plain) community in the U.S. Midwest. Navigate NBS staff worked with families from the clinic’s community, providers, and clinic staff to create a prenatal education flip book and recruited NBS partners to share and disseminate the flipbook. 

During routine, third-trimester visits, clinic staff invited expectant mothers to participate in the educational initiative by giving them an informational card, which included a QR code directing them to the digital flip book, and/or providing a paper copy of the flipbook. The NBS Prenatal Education Initiative was approved as QI Project No. 2021-1018 under the Harris Health IRB.

Participants at both sites were asked five questions both prior to reviewing the flip book (pretest) and after their review (posttest). These questions were designed to have one right answer and test participants’ knowledge of NBS. Most questions on the pretest and posttest were true or false. We shared correct answers with participants at the end of the posttest. To assess participants’ understanding of the definition of NBS, the tests presented a series of options, including different time periods and types of testing.

In R statistical software (Version 2021.09.0 Build 351), we used a series of Wilcoxon signed ranked tests for nonparametric data to assess pre- and posttest differences for each item. We calculated a total knowledge score for pre- and posttests by summing the number of correct responses, and then conducted a dependent-samples *t*-test to determine statistical significance. All tests used an α = 0.05 as a threshold to assess statistical significance. 

### 2.3. Strategy 3: Social Media Campaign

Social media is a frequently used tool for communicating preconception and pregnancy related health topics with over 90% of adults ages 18–24 engaging with some social media platform [19]. To increase awareness of newborn screening in medically underserved communities, the program initiated and evaluated the efficacy of paid social media campaigns to engage and educate families. An initial 1-month pilot was facilitated in 2021 and showed that this strategy shows promise at reaching individuals in MUAs with NBS messages. 

To expand reach and to further assess effectiveness of using social media platforms to educate families about newborn screening, Navigate NBS staff tested a 6-month paid awareness campaign on Facebook and Instagram, which are commonly used platforms. Social media ads were first developed, in English and Spanish, to support the outreach and engagement from diverse communities. Ads were created with input from family and industry experts using culturally diverse images, plain language, clear calls to action, and a mixture of static (still images) and dynamic (video/animated) graphics. Geotargeting parameters were defined and enabled directed outreach to specific U.S.-based ZIP codes aligning with MUAs as identified through the Health Resources and Services Administration, the funding agency for this project [20,21]. Through this campaign, the program was able to direct and optimize viewing of NBS messages to specific individuals who met one of the following audience parameters: Women between the ages of 18–44;Those who identify as expecting parents;Those interested in maternity, pregnancy, and prenatal care;Individuals with a household income of USD <30,000/year; andThose who live in one of the ZIP codes identified as an MUA by HRSA

No additional demographic information was collected. Priority populations in the campaign were shown educational messages and images about newborn screening and were prompted to click a link directing them to an online educational “flip book” containing more information about newborn screening.

The awareness campaigns measured the number of impressions (i.e., number of times a campaign ad was seen), clicks (i.e., number of interactions with a campaign ad such as likes, dislikes, comments), and link clicks (i.e., number of individual who clicked on the link in the social media ads and landed on the prenatal flip book), and the click through rate [CTR] measured the number of users who clicked on a link compared to the total impressions. 

## 3. Results

### 3.1. Strategy 1: Online Education Modules

The initial online training curriculum, Navigate Newborn Screening, for English-speaking families, launched in February 2020. From launch through June 2023, 719 people registered for the online modules, including 307 parents and families and 412 individuals from secondary audiences, such as students, providers, and other professionals. A course completion was documented when individuals viewed and completed learning checkpoints for Section 1: The Newborn Screening Process; Section 2: Newborn Screening Results; Section 3: Types of Conditions Detected; and completed an evaluation form at the end of Section 3. In total, 138 parents and families completed the minimum required sections (Sections 1–3) or beyond and the evaluation, including 110 who completed the English modules and 28 who completed the Spanish modules. In addition, 191 individuals from secondary audiences completed the modules. 

Of the 138 family participants that completed at least three modules, 129 (93.5%) said they agreed or strongly agreed with the statement: After taking the educational module, I feel more knowledgeable about newborn screening. 

Notably, the online modules also contributed to participant reports of increased self-efficacy to perform behaviors related to NBS. Of the 138 family participants, 125 (90.6%) reported increased confidence to find information about NBS, 124 (89.9%) reported increased confidence to talk to their healthcare provider about NBS, 117 (84.8%) reported increased confidence to advocate for their health or their child’s health, and 112 (81.2%) reported increased confidence to serve as a leader in the NBS community (Figure 1). 

### 3.2. Strategy 2: Prenatal Education

Over a 10-week period, a total of 83 participants completed the initiative. In the high-risk obstetric clinic-based community (Pilot Site 1), 143 participants interacted with the pilot materials and 56 (39%) completed all phases of the pilot (i.e., pretest, flipbook review, posttest). Approximately 90% of the 56 participants completed the surveys digitally and in their preferred language (English or Spanish). In the Plain community (Pilot Site 2), 27 participants completed the survey using printed/paper materials and in English, which reflected the communication preferences of these communities. It was unknown how many received a flipbook overall. The average time spent reading the flip book was 2.5 min. The pre and post tests were designed to assess the extent to which reading the flip book during a clinic visit increased knowledge and self-efficacy around NBS. (see results in Table 1). Results indicated that the flip book significantly increased participants’ awareness of NBS, knowledge of NBS, and confidence to perform NBS-related behaviors.

#### 3.2.1. Assessing Knowledge

The NBS Prenatal Education Initiative improved knowledge about NBS among expectant parents in both sites. On average, in Pilot Site 1, participants answered 48.7% correct during the pretest (1−2 items) and 91.2% correct during the posttest (4−5 items), *t*(55) = −9.90, *p* < 0.001. In Pilot Site 2, participants answered 66.7% of the pretest questions correct (3–4 items) and 98.5% of the posttest questions correct (5 items), *t*(26) = −6.98, *p* < 0.001. Table 1 shows the number of participants selecting the correct answer at pretest and posttest for both Pilot Sites.

#### 3.2.2. Assessing Self-Efficacy and Confidence

The NBS Prenatal Education Initiative also improved self-efficacy among expectant parents in both sites. In Pilot Site 1, 23 participants (41.1%) reported they had heard of NBS and 52 agreed or strongly agreed that it was important (92.9%) prior to reviewing the flip book. After reviewing the flip book, 89.3% of participants indicated they agreed or strongly agreed that they knew more about NBS (*n* = 50), 94.6% reported that knowing about NBS is important (*n* = 53), 96.4% trusted the information provided (*n* = 54), 94.6% felt capable talking to their doctor about NBS (*n* = 53), and 92.9% knew where to look for more information (*n* = 52).

In Pilot Site 2, all 27 participants (100%) indicated they agreed or strongly agreed that they knew more about NBS, reported that knowing about NBS is important, trusted the information provided, felt capable talking to their doctor about NBS, and knew where to look for more information after reviewing the flip book.

### 3.3. Strategy 3: Social Media Awareness Campaigns

The initial 30-day pilot social media campaign ran from 8 August 2021 to 9 September 2021 and generated 1.99 million impressions. The initial goal to reach 30% of the intended audience with NBS ads was exceeded when 67% of the available audience was reached. This notable reach of the campaign demonstrated the ability of digital outreach to reach individuals in medically underserved areas with information about NBS.

The 6-month paid campaign on Facebook and Instagram ran from 1 March 2022 to 31 August 2022 and generated 3.42 million impressions. The campaign was able to reach 2,851,115 unique viewers with a frequency of 1.2 ad views per person, which was 52% of the available audience (4.5–6.5 million people). Of those who saw the ads, 14,570 individuals clicked through to the Navigate NBS flip book. Early in the campaign, text-only ads generated more engagement and clicks than those with images; however, over time, ads with images of babies generated the highest volume of engagement. Additionally, ad colors appeared to impact engagement, with ads using orange backgrounds generating more engagement than ads using a teal background. Figure 2 shows the educational ads used in the social media campaign. 

This campaign was able to achieve a link CTR of 0.43% (14,570 link clicks/3,420,000 impressions) which is lower than health-related benchmarks (0.83%) but similar to the Facebook average (0.40%). Of note, once an individual exited the Facebook or Instagram platforms by clicking into the prenatal flip book, engagement decreased. Approximately 23.6% of users engaged with the flip book beyond the initial link click (1041 flip book views/4405 impressions). 

## 4. Discussion

Over the course of a 5-year period, the *Navigate NBS* program was able to test a variety of strategies to reach and educate parents about NBS, including those living in MUAs. We described three promising strategies, including online learning modules, a prenatal education pilot program, and a social media awareness campaign.

The online modules demonstrated self-reported increases in both knowledge and self-efficacy around NBS. Notably, while developed primarily for families, more than half of enrollments were from secondary audiences, suggesting that there is interest in these modules from diverse audiences and that both providers and families may benefit from this type of education. Through the two pilot sites in the prenatal education initiative, we demonstrated that a clinic-based initiative for medically underserved women can increase knowledge about NBS. Finally, the results of both the 30-day and 6-month social media campaigns demonstrated that social media is a promising tool to reach a large number of diverse and often hard-to-reach families with short, specific educational messages. These campaigns demonstrated promise in raising awareness and knowledge of NBS for families, particularly in medically underserved communities. 

The three primary strategies used in the Navigate Newborn Screening Program demonstrated the need for a multifaceted approach to meet the needs of all families. Additionally, these strategies provide options for those responsible for providing families with information but lacking appropriate resources to complete large scale implementation projects. The online module provided in-depth training and education to families and increased knowledge and self-efficacy in important ways, but it may be limiting in its ability to reach large audiences, especially those limited by technology. Additionally, there was a higher-than-expected level of interest from secondary audiences suggesting both a need and opportunity for additional training for both providers and families. The clinic-based prenatal initiative was effective at increasing knowledge and self-efficacy for expectant mothers in MUAs at a critical time in the prenatal period. This initiative responded to a critical gap [14] by providing clinics with educational tools to explain NBS in clear and understandable ways during prenatal visits. The success of this strategy is notable; however, the process of engaging clinics and integrating the flip book into the clinic workflow is resource-intensive and relies on a clinic-champion to fully succeed. Finally, the social media awareness campaign substantially increased the scale of the program, reaching millions of expecting parents in MUAs; however, the depth of knowledge gained may be more limited. Each strategy had its strength in reaching families, and when combined, demonstrated the power of a comprehensive program to engage families and meet them where they are.

As the prenatal and early childhood landscape is increasingly complicated with testing options and complexity discerning accurate information, it is important for NBS information to be trusted, relevant, and fit into the lives of those we want to reach. Engaging expectant families, especially those from historically underserved communities, is necessary to increase knowledge and confidence which leads to overall improved outcomes for families [5,9]. Barriers to family engagement expand far beyond healthcare systems and newborn screening, and include a multitude of influences that challenge how families collaborate, communicate, and make decisions. Insufficient communication channels; ineffective outreach; language and cultural differences; disparities to access; and relevant, real-time information are just a few of the well-described barriers that impact how families engage in systems of care [16]. 

This program provided the opportunity to generate important lessons for the future of NBS education. 

**A Recommended Model for Successful Family Engagement:** Successful education must be multifaceted and requires multiple strategies to meet the diverse and nuanced needs of families along their newborn screening journey. The Navigate Newborn Screening Program, led by Expecting Health, implemented an array of strategies and resources to assess and improve knowledge with families in the newborn screening system. Improved knowledge correlates with increased confidence to engage with the health system including a higher likelihood to ask their health provider questions, locate needed information when they need it and advocate for themselves and/or their family. Through the development and implementation of innovative strategies, practical resources and the multifaceted approaches to education described in this work, more families are more aware, educated, and confident about newborn screening. By utilizing quality improvement methodology for improvement process design and collaboration with families as partners, this work highlights several elements leading to success. 

**Inclusive Outreach**: Engagement methods are designed to reach a diverse range of potential partners, including individuals from underrepresented communities, marginalized groups, and diverse backgrounds. This involves prioritizing populations for outreach efforts, utilizing culturally appropriate communication channels, and collaborating with community organizations to ensure that diverse voices are included in the research process. Digital methodologies such as online social media campaigns successfully expand reach, particularly to communities that are historically underserved or more difficult to reach. Although women were the primary target of these strategies, future scaling and inclusive outreach can target expectant fathers and people of all genders.

**Co-creation and Co-design**: Incorporating a broader range of experiences and knowledge into the design and co-creation of the trainings, prenatal educational book and online social media campaign created more ways for families to be involved, identified real needs, and incorporated practical approaches for educating a broader group of families, which led to higher levels of engagement overall. Engaging families as partners and stakeholders from the beginning of the project in the development and refinement of methods helps ensure that the process is inclusive and relevant to the needs and perspectives of various communities.

**Relevance and Timeliness:** Current practices indicate that most families first learn about newborn screening at the time of screening, at the time of results, or not at all. The assessments completed through the online training modules and prenatal education efforts suggest that providing opportunities for families to access this information earlier in their health care journey improves awareness, knowledge, and confidence. 

**Ongoing Evaluation and Reflection**: It is essential to continuously evaluate and reflect upon engagement methods to ensure they are achieving their intended goals of promoting diversity and inclusion. Researchers should gather feedback from partners, assess the effectiveness of the methods used, and make necessary adjustments to better support the diversity of those involved in the research process.

**Innovate and Scalable Methods**: Education tools must be innovative and engaging, but simple to implement. Through continuous quality improvement methodologies, we were able to identify opportunities for improvement, test them and scale many of our activities to expand the priority populations we were able to reach particularly those in medically underserved communities.

By incorporating these considerations into the development, implementation, refinement, and evaluation of education and engagement methods, other researchers can create more inclusive processes that embrace diverse perspectives, foster collaboration, and produce more comprehensive and impactful outcomes. The strategies described here can be applied broadly and for other topic areas beyond newborn screening. Additional studies are needed to determine the impacts of education on NBS health outcomes. With these data and the work of others, the NBS community is poised to tackle scaling work like this to reach more families.

## Figures and Tables

**Figure 1 IJNS-10-00044-f001:**
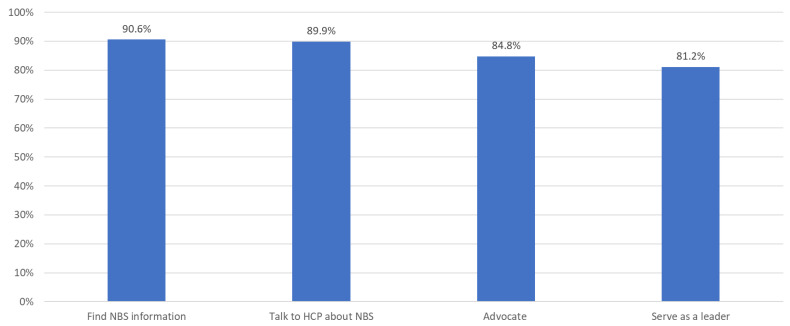
The percentage of family participants who reported increased confidence to perform NBS-related behaviors after completing the online modules.

**Figure 2 IJNS-10-00044-f002:**
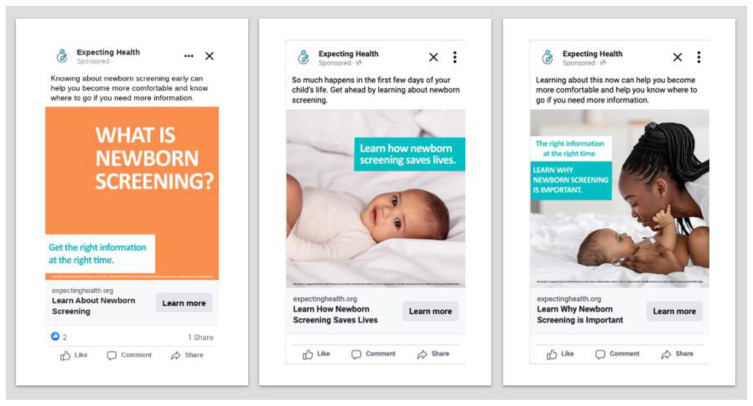
Example social media ads Not for distribution without permission from Expecting Health.

**Table 1 IJNS-10-00044-t001:** Pre- and posttest knowledge comparisons for two Pilot Sites.

	Pilot Site 1 (*n* = 56) Participants Selecting the Correct Response *n* (%)		Pilot Site 2 (*n* = 27) Participants Selecting the Correct Response *n* (%)	
Knowledge Assessment Questions	Pretest	Posttest	Pretest	Posttest
What is newborn screening? (Correct response: Tests performed 24 to 48 h after birth)	18 (32.1%)	48 (85.7%) **	16 (32.1%)	27 (100.0%) *
There are 3 parts to newborn screening. (Correct response: True)	25 (44.6%)	54 (96.4%) **	9 (33.3%)	27 (100.0%) *
Newborn screening helps identify babies who may be at risk of having serious health issues. (Correct response: True)	47 (83.9%)	56 (100.0%) *	43 (85.2%)	27 (100.0%) *
Newborn screening usually takes place one week after a baby is born. (Correct response: False)	22 (39.3%)	49 (87.5%)	21 (77.8%)	27 (100.0%)
An abnormal result always means there is something wrong with my baby. (Correct response: False)	9 (72.0%) ^a^	46 (82.1%)	21 (77.8%)	25 (92.6%) *

* Difference between pretest and posttest is statistically significant at *p* < 0.01 (** *p* < 0.001) using a Wilcoxon signed rank test for nonparametric data. ^a^ Item was not asked during the first few weeks of pilot, *n* = 25.

## Data Availability

The raw data supporting the conclusions of this article will be made available by the authors on request.

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
