# Peer review of "One Size Does Not Fit All: A Multifaceted Approach to Educate Families about Newborn Screening"

_2409-515X, 2024, doi:10.3390/ijns10030044_

Round 1

Reviewer 1 Report

Comments and Suggestions for Authors

This is an interesting paper testing educational approaches for newborn screening.    The authors are leaders in the newborn screening educational field, with a lot of experience in family engagement and education.   The rationale for their work is well described and important. 

Comments:  

1.      The model for improvement figure does not help the paper, and it is described in the text itself.   A better figure would be how this model applied to this study.  What did YOU do.   I would suggest that you take the first paragraph after the figure and make it into a new figure.  

2.      It would be helpful if there were more details in the methods about hos the educational materials were developed.   What sources were used, what content was incorporated, how did the authors decide what was important to  include?    Was there expert feedback on the content before it was tested by families.  

3.      In the methods, page 3 line 107 the authors list results from their assessment.  Those numbers belong in the results section.   They also state there is a difference between them   You can’t state ‘differences’ without statistical testing.   Further, the authors go on to speculate what those results mean, which best fits in the discussion.  

4.      THe methods should describe the statistical testing that was performed.  It is listed under strategy 2 in the results.   This needs to be in the methods.

5.      All the references need to be formatted similarly.   Most are Last name, first name, some are  first name, last name.   The references to websites need to include date accessed, last date updated.   

6.      Need to do statistical testing on all quantitative results where you state there are (or are not) differences.  Several places state that values are different or not different with no testing performed.   

7.      Very minor point: Strategy 3:   saying social media is a well-known tool seems awkward.   Many frequently used tool?

8.      In the results, Figure 2 caption needs to state that this was in family participants (not the others).  I think that is correct?  

9.      The social media campaign – if you are going to state that images with babies were more likely to be clicked, you need to present that data.   Same for the colors.    And do statistical testing.  

10. I do not think 0.43% is different than the facebook average.    You could test this, or just say ‘similar to’   

11. The final paragraph in the results is really a discussion point.    IN the results section you should only list results, not interpretation of the results.  

12. Discussion – in the first paragraph, I would provide a reference to how you know that the landscape is more complex. 

13. This is a style point, but I would put your main points at the beginning of the discussion.   What did YOU find, vs. what is already known in the literature?

14. Some of the discussion points may be overstated/not necessarily supported by the data.   For example, “This initiative filled a critical gap”  - I don’t think you measured gaps or determined if the gaps were filled.    It provides a good resource for them, but the authors need to be careful not to state conclusions beyond what their data supports. 

15. In the discussion section, the authors state , “…the social media campaign dramatically increase the scale…. However the depth of the knowledge may be more limited.”     Style point – I would delete dramatically as that is subjective, and state that while the depth of knowledge gained may be limited, you weren’t able to measure it in this study. 

Reviewer 2 Report

Comments and Suggestions for Authors

I applaud any initiative to improve any educational initiative related to newborn screening, as this is crucial to results in qualitative consent procedures. I’m not sure that there is sufficient evidence that these examples are indeed ‘effective’, but these rather read as potential approaches, so that I suggest the authors to reconsider the ‘framing’ of this paper.

Specific comments

Line 33: this success is not restricted to the US, so suggest to rephrase. Alternatively, it should be clearer that you have your focus on the US setting.

Line 35: do you really mean exacerbated ?

Line 38: ‘benefits and positive outcomes’= does this refer to the consent procedure, to the second line controls in the event of an abnormal result, or does this refer to the refusal ?

Line 44: at delivery, or is this (also) to be done during pregnancy (you mention this lower) ?

I understand that the Navigate Newborn Screening has the intention to reach the ‘underserved’ or ‘underrepresented’ populations, but is there also evidence that this target is reached to a relevant extend ? At first glance, all efforts need online access and skills. Is this intentional, and have you considered the circumstances of the targeted populations (online access, literacy beyond language) into account ?

Second, the tools seems to target women ? Were fathers or partners been targeted/considered ?

We need some more information on what you mean with social media ? facebook tiktok, or what tools have been applied, and why/how have these tools been selected (these are mentioned in the results section) ?

Related to the 3.1 section, is it known ‘who’ attended (the characteristics of the target population ?)

Related to the 3.2 section, do we know how many (%) were invited versus the number of women that took the course (acceptance rate).

Reviewer 3 Report

Comments and Suggestions for Authors

The authors describe an important multi-dimension project to benefit families that has timely relevance extending to the broader newborn screening community. Comments are relatively minor; most notable regards enhancing references and corresponding citations, especially in the introduction. 

This reviewer welcomes re-reviewing if modification are made to the original manuscript and if requested by the editors.

Please note, this reviewer had not verified data, including statistical analysis

From the perspective of this reviewer, please consider consistently italicizing the project name (Navigating NBS) throughout. Readability was enhances when authors occasionally did so, for example  in lines 74-75, 140,  204

From the perspective of this reviewer, please consider capitalizing “site” whenever noting in text alongside Pilot – “Pilot site1”  or Pilot site 2. 

From the perspective of this reviewer, the cited literature (reference list) is overly sparse in the introduction and would greatly benefit from more inclusion of the work of others, with some historical content that supports the importance for this effort and, especially, more current projects of relevance, and particularly in underserved populations.

Line 5: “and” is  misplaced

Line 11: From the perspective of this reviewer, “appropriate” is too vague a term and value-laden without explanation, especially to begin an abstract or early introduction

Line 31- 33: Please provide citation  

Line 33- 36: Please provide citation

Line 42: From the perspective of this reviewer, “should” either needs citation or an explanatory clause such as ‘from the authors’ experience” etc

Lines 141-144: From the perspective of this reviewer, these important details would benefit by separating. Please consider reformatting , such as bulleting, table, etc.

Line 191: Secondary audience: Was any demographics collected?

Line 308: From the perspective of this reviewer, “bombardment” may be perceived as value-laden and overly encompassing by some readers. Please consider tempering, with word substitution or addition of a modifier. 

Please also clarify/address, the contradiction between the authors’ perspective of “bombardment” with the longstanding critiques that many feel there is a lack of these resources. 

Line 330: Please clarify authors’ use of “success” , one example might be “based on self-reported…” --referring back to “self-reported” in line 218.

Round 2

Reviewer 1 Report

Comments and Suggestions for Authors

Thank you for your careful revision of the manuscript.   You have addressed my concerns.  

Author Response

Thank you.  

Reviewer 2 Report

Comments and Suggestions for Authors

the revised version is in line with the suggestions provided

Author Response

Thank you very much for your support of this manuscript.  We sincerely appreciate your time, attention and comments. 

Reviewer 3 Report

Comments and Suggestions for Authors

Thank you to the authors for contributing this research and manuscript to advance the field. Your efforts addressing reviewers' comments enhanced the manuscript significantly. One point of consideration remains in that citation 2 (NBS and IEMs review article) dates back to 1997. From this reviewer's perspective, the addition of a more recent review on this topic is suggested. One potential article is 

Arnold G. Newborn Screening for Inborn Errors of Metabolism: Review. OBM Genetics 2023 Volume 7, Issue 4 doi:10.21926/obm.genet.2304197

ALL ELSE IS VERY STRONG--

Author Response

Thank you for sharing this citation. We agree that numerous citations are old and this more current review has been added in as a reference.  Thank you very much for sharing. 

We sincerely appreciate your time and attention to supporting this manuscript.